

# Pacing during an ultramarathon running event in hilly terrain

Hugo A. Kerhervé[1], Tom Cole-Hunter[2,3], Aaron N. Wiegand[4] and Colin Solomon[1]

[1] School of Health and Sport Sciences, University of the Sunshine Coast, Sippy Downs, QLD, Australia
[2] ISGlobal, Barcelona Institute for Global Health, Barcelona, Spain
[3] Department of Environmental and Radiological Health Sciences, Colorado State University, Fort Collins, CO, United States
[4] School of Science and Engineering, University of the Sunshine Coast, Sippy Downs, QLD, Australia

Corresponding author
Hugo A. Kerhervé,
hkerherv@usc.edu.au

## ABSTRACT

**Purpose**. The dynamics of speed selection as a function of distance, or pacing, are used in recreational, competitive, and scientific research situations as an indirect measure of the psycho-physiological status of an individual. The purpose of this study was to determine pacing on level, uphill and downhill sections of participants in a long (>80 km) ultramarathon performed on trails in hilly terrain.

**Methods**. Fifteen ultramarathon runners competed in a 173 km event (five finished at 103 km) carrying a Global-Positioning System (GPS) device. Using the GPS data, we determined the speed, relative to average total speed, in level (LEV), uphill (UH) and downhill (DH) gradient categories as a function of total distance, as well as the correlation between overall performance and speed variability, speed loss, and total time stopped.

**Results**. There were no significant differences in normality, variances or means in the relative speed in 173-km and 103-km participants. Relative speed decreased in LEV, UH and DH. The main component of speed loss occurred between 5% and 50% of the event distance in LEV, and between 5% and 95% in UH and DH. There were no significant correlations between overall performance and speed loss, the variability of speed, or total time stopped.

**Conclusions**. Positive pacing was observed at all gradients, with the main component of speed loss occurring earlier (mixed pacing) in LEV compared to UH and DH. A speed reserve (increased speed in the last section) was observed in LEV and UH. The decrease in speed and variability of speed were more important in LEV and DH than in UH. The absence of a significant correlation between overall performance and descriptors of pacing is novel and indicates that pacing in ultramarathons in trails and hilly terrain differs to other types of running events.

## INTRODUCTION

The dynamics of speed during self-paced locomotor exercise, or pacing, are used in recreational, competitive and scientific settings as an indicator of exercise intensity
(*Abbiss & Laursen, 2008*), and fatigue (*Knicker et al., 2011*). Three general types of pacing (negative, even, positive) are commonly identified in the analysis of running performance, using the direction of the changes in time per km or speed, as a function of distance or exercise duration. Events longer than the marathon (ultramarathon, UM) are increasingly popular in recreational and competitive settings and are increasing being used in scientific research. However, there is a relative paucity of systematic descriptions of pacing in long (>80 km) UM, especially those performed on trails and in hilly or mountainous terrain, which could be due to the difficulties associated with the monitoring of individual participants in remote areas and over long durations.

Positive pacing (decreasing speed) and a subset of positive pacing including a final end-spurt referred to as parabolic pacing (*Abbiss & Laursen, 2008*), were observed in UM, such as during a short (45 km) trail UM in recreational runners (*Angus & Waterhouse, 2011*), a 100 km event on a level, multi-loop course in elite runners (*Lambert et al., 2004*), a 105 km mountain trail UM in competitive runners (*Kerhervé, Millet & Solomon, 2015*), and a 161 km mountain trail UM in the five fastest runners over a 28 year period (*Hoffman, 2014*). Positive pacing has also been observed in other forms of ultra-endurance exercise, such as a 24 h treadmill run (*Gimenez et al., 2013*), or during an ultra-endurance triathlon event consisting of ten consecutive Ironman distance triathlons (10 × 3.8 km swimming, 180 km cycling, 42 km running) in 10 days (*Herbst et al., 2011*). Pacing is also characterised by the magnitude of speed loss and the variability of speed (using the coefficient of variation of speed), which were found to be lower in faster compared to slower participants during UM running events (*Lambert et al., 2004*; *Hoffman, 2014*), in agreement with what has been observed in events up to the marathon distance (*Ely et al., 2008*; *Haney & Mercer, 2011*).

In contrast to these findings, we measured in a previous study on trained UM runners a higher magnitude of speed loss in faster compared to slower runners, no significant relationship between the variability of speed and performance level, and a novel significant negative relationship between the total time stopped and performance level, in a long mountain UM (*Kerhervé, Millet & Solomon, 2015*). Additionally, speed on level, uphill and downhill sections increased in the last 10% of the event. These results indicate that pacing may have been regulated conservatively in anticipation of topographic difficulties, and that faster runners paced less conservatively than slower runners. Overall, the literature specific to UM and ultra-endurance exercise has highlighted the protective nature of fatigue in situations where the physical integrity of participants could be compromised, such as greater neuromuscular (*Millet et al., 2011b*) or biomechanical alterations (*Morin et al., 2011b*) in events with large elevation gain and loss compared to UM events on level ground (*Martin et al., 2010*; *Morin, Samozino & Millet, 2011a*).

Therefore, the aim of this study was to record the individual pacing of participants in a long trail UM in hilly terrain, in order (i) to measure the dynamics of speed selection as a function of terrain (level, uphill and downhill sections), and (ii) to further investigate the relationship between performance and pacing characteristics (speed loss, speed variability, total time stopped). Speed was expected to decrease in all participants and at all gradients as a function of the distance and duration of the event.

## METHODS

This study was approved by the university research ethics committee (Queensland University of Technology, project 0900001233). The study participants were recruited using advertisements on a specialised forum and researchers networks, from individuals already registered to compete in the Great North Walk 100s (NSW, Australia), a long (~173 km) and hilly UM running event including 6 checkpoints and a total elevation gain and loss of approximately 3,000 m. This event offered the opportunity to receive an official classification and time for participants failing to complete the entire event if they reached the 4th checkpoint at ~103 km, which we included in the study. Both distances are presented in this article as 173-km and 103-km. After providing written informed consent, 19 participants were equipped with a commercially-available Global Positioning System (GPS) device (BT-Q1000; Qstarz International, Taipei, Taiwan) fitted to their clothing or pack.

The GPS devices used in this study were selected for their light weight (~100 g including the battery) and long battery life (tested to record for more than 40 continuous hours at a sampling rate of 0.2 Hz). The accuracy of measures of geographical position and speed were tested following published procedures (*Townshend, Worringham & Stewart, 2008*). Positional accuracy was found to be within the range provided by the manufacturer (100% of measures within 3 m of a known geodetic survey landmark, 84.5% of observations measured within 2 m, and had a mean distance of $1.57 \pm 0.43$ m). The calculated velocity (see following section) was found to be in excellent agreement (95% limits of agreement = $0.24 \pm 0.12$, typical error expressed as a % of CV = 1.2, standardized error of the estimate = 0.03 km h$^{-1}$) and perfectly correlated to speed determined using chronometry over a known distance (100 m) over 50 trials ($r = 1.00$, $p < 0.001$).

### Distance and Speed

The distance between points at the surface of a sphere can be calculated using simple spherical trigonometry. However, the earth is not a sphere, but an oblate spheroid akin to an ellipsoid with the following dimensions: equatorial radius $\approx$6,378.13 km, polar radius $\approx$6,356.752 314 245 km and flattening f $\approx$1/298.257223563 (*Defense Mapping Agency, 1990*). The calculation of point to point distances at the surface of an ellipsoid can be improved compared to spherical trigonometry using the inverse Vincenty formulae (*Vincenty, 1975*). The point-to-point distances were obtained from an internet-based utility (GPS Visualizer; http://www.gpsvisualizer.com) using geographical positions (latitude and longitude). We found the distances obtained were in exact agreement ($r = 1.00$, $p < 0.001$) with our preliminary measures of point-to-point distances using the Vincenty formulae performed on 10 data sets. Therefore, we used the internet-based calculations of the formulae as a simple and generalisable procedure to obtain point-to-point distances.

Point-to-point speed was subsequently calculated using the ratio of point-to-point distances and time (one data point every 5 s) between each datum. Preliminary calculations revealed that GPS devices did not discriminate for speeds slower than 1 km h$^{-1}$ (0.28 m s$^{-1}$ or 1.39 m in 5 s) based on the typical error in speed in a static position (drift, when a device will record speed values due to the non geo-synchronous nature of the constellation of satellites). At the other end of the speed spectrum, it was considered

that speeds higher than 20 km h$^{-1}$ (5.56 m s$^{-1}$ or 27.8 m in 5 s) were not expected during a long UM and originated in signal jamming (which can occur due to the signal from a satellite being too weak which forces the ground based receiver to pair to another satellite). These erroneous distance and speed data were assigned a value of zero, and all speed values were then smoothed in order to further increase the signal-to-noise ratio. For smoothing, a 9-pt weighted average was graphically compared with 3-pt and 15-pt weighted averages and considered satisfactory as it provided a balanced sensitivity to individual observations for slow and high speeds. This procedure limited the effect of signal drift and jamming (higher distance and speed due to erroneous values). These procedures were sensitive to periods of zero speed values, which corresponded to the location, via expected relative distances, of checkpoints in the race.

## Elevation changes

We used previously published procedures (*Kerhervé, Millet & Solomon, 2015*) to alleviate the inaccuracies in GPS-based elevation (*Townshend, Worringham & Stewart, 2008*). The online utility (GPS Visualizer; http://www.gpsvisualizer.com) was used to match the recorded geographical positions (determined by latitude and longitude) to a DEM elevation datum from the National Aeronautics and Space Administration (NASA) Shuttle Radar Topography Mission (SRTM) database. As the resolution of DEM is relatively low for the study of human locomotion (90 m, only the SRTM3 was available where the current study was conducted), the procedure can create a series of steps when increasing or decreasing elevation every time a contour line is crossed. Likewise, it can also mean that actual changes in elevation are not detected when data points were recorded inside the same contour line. In the absence of an existing method to reconstruct accurate elevation data, we applied the same 9-point weighted average smoothing procedure we used in the speed data to calculate more realistic elevation data. Gradient was then calculated as the change in elevation divided by the horizontal distance between two points.

## Variables and statistical analysis

The identification of relevant data was performed for each participant using official race results, GPS time-stamps and variations of speed (increase or decrease) indicating changes in position at the start and finish lines. Speed and gradient values were computed as a function of relative distance for each participant, where total individual distance represents 100% of the distance completed for the 173-km, and 60% for the 103-km event participants. Relative distance was used instead of the actual distance values because of the difference in total distances across participants. The dynamics of speed were determined relative to each participant's average speed over the entire event (100%) in order to increase the relevance of inter-individual comparisons. The mean relative speed values corresponding to level (LEV; −2.5 to 2.5% gradient), uphill (UH; 2.5–100% gradient), and downhill (DH; −100 to −2.5% gradient) were computed in sections of 5% of the total distance completed, which ensured a sufficient amount of data points in each section in the three gradient categories. The minimal amount of data points within one section occurred at 25% of total distance in the level gradient category (64 observations for one participant, which corresponded

to 2.76 km for the participant), for an average number of observations in each gradient category of $360 \pm 98$ (LEV), $439 \pm 60$ (UH) and $386 \pm 67$ (DH).

We initially assessed the normality (Kolmogorov–Smirnov test) and homogeneity of variances (Fisher's F test) of the 173-km and 103-km participants for LEV, UH, DH, and overall relative speeds. We then used independent t-tests to evaluate whether differences in relative speed existed between the 173-km and 103-km participants in the same categories. Due to varying lengths of data sets (173-km and 103-km distances), we investigated the dynamics of speed within each gradient category using a one-way ANOVA on ranks (Kruskal–Wallis test) and a pairwise multiple comparison (Dunn's method) when required, to determine and locate potentially significant differences between sections of relative distance.

The relationship between the level of performance (individual average speed) and the variability of speed (coefficient of variation of speed), the magnitude of speed loss (slope of the linear regression of speed over the entire event) and the total time stopped (assumed to correspond to resting, eating, clothing and gear change, toilet, other), were assessed using correlations. Assumptions of normality were first tested using a Shapiro–Wilk test, and Pearson's product-moment correlation was used to calculate each relationship.

All statistical analyses were performed using the computing program and its associated packages R (*R Development Core Team, 2015*). The level of significance was set at $p < 0.05$.

## RESULTS

The datasets from four participants were not included in the analysis, due to either not completing the event, or discrepancies between official results and GPS data (difference in finish time greater than 5%). A total of 15 GPS datasets were used for analysis (see Fig. 1): ten participants completed the 173-km distance averaging $32.9 \pm 3.6$ h (range 26.5–36.3 h), with a mean total distance of $173.0 \pm 4.0$ km ($5.68 \pm 1.51$ km h$^{-1}$), and five participants completed the 103-km distance averaging $18.9 \pm 2.3$ h (range 15.6–21.1 h) with a mean total distance of $101.9 \pm 2.3$ km ($5.91 \pm 1.48$ km h$^{-1}$). Weather conditions on race day were dry (no precipitations), the temperatures ranged 14.9–22.8 °C at the start point, 10.2–29.1 °C at the 103-km checkpoint, and 11.4–26.8 °C at the finish line, respectively.

The initial testing for normality and homogeneity of variances did not reveal any significant differences between 173-km and 103-km participants. There were no significant differences in means in the relative speed of the 173-km and 103-km participants for all categories of gradients (overall: $F = -1.38, p = 0.17$; LEV: $F = -0.99, p = 0.32$; UH: $-1.37, p = 0.18$; DH: $F = -0.27, p = 0.79$). Therefore, all remaining analysis include the 173-km and 103-km participants.

Positive pacing was observed in all participants and in all gradient categories, except in one participant in negative gradients. The mean decrease in speed was $-4.35 \pm 3.0$ km h$^{-1}$ (LEV), $-2.26 \pm 1.4$ h$^{-1}$ (UH), and $-4.36 \pm 2.2$ h$^{-1}$ (DH). Relative speed was the most variable in LEV (coefficient of variation: 0.32) and reached a minimum at 50% of total distance (significantly lower than all observations between 5% and 40%). The relative speed increased from 50% to 60%, 85%, 90%, and 100%, and did not significantly decrease between any sections before 50% and 100% (Fig. 2A). Relative speed was the least variable
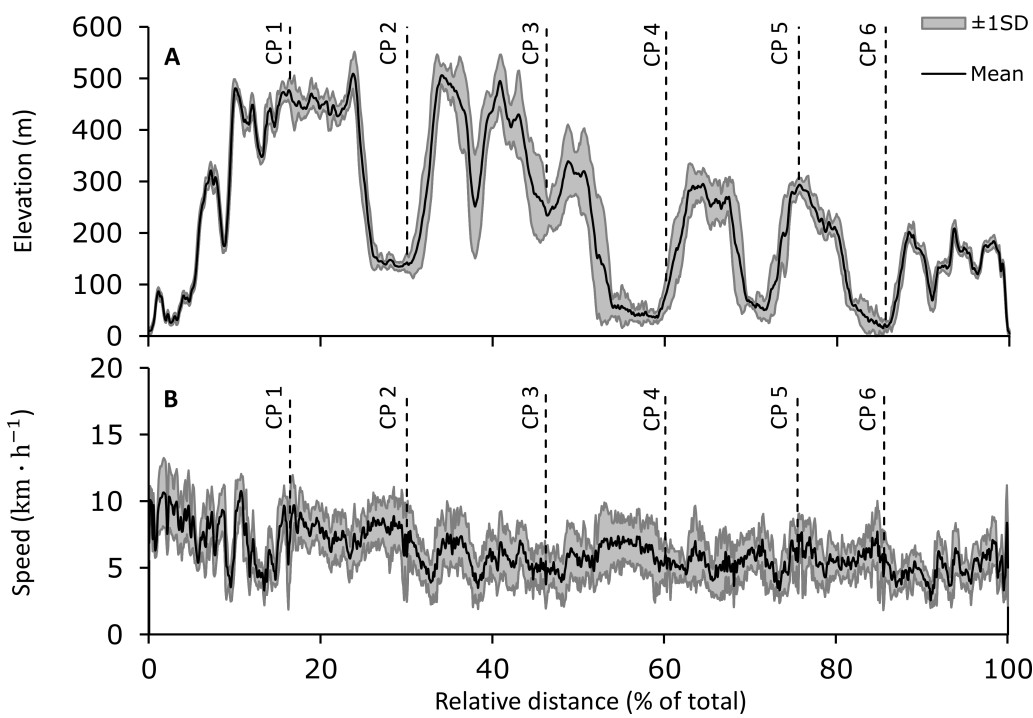

**Figure 1 Elevation and speed.** (A) Mean (SD) group DEM data from geographical positions. (B) Mean (±SD) group calculated speed. Abbreviation: CP are official race checkpoints.

in UH (coefficient of variation: 0.22) and reached a minimum at 95% of total distance. Relative speed increased from 35%, 40%, 45% and 50% to 60%. Relative speed increased from 95% to 100%, which was not different to any other observation (Fig. 2B). Relative speed in DH reached a minimum at 95% of total distance (coefficient of variation: 0.26). The relative speed at 20% increased compared to 15%, but no other significant increase was observed. The relative speed at 60% was not significantly different to any other observation, and the relative speed at 100% was significantly lower than 5%, 10%, 15%, 20%, 30% and 35% (Fig. 2C).

There were no significant correlations between overall performance and variables of pacing (Table 1). The dynamics of total time stopped as a function of relative distance are presented in Fig. 3.

## DISCUSSION

In this study, we collected and reported the longest systematic description of pacing of runners in a long, hilly UM running event, using a method that created no disturbances to normal running event situations.

The primary finding of this study was that positive pacing (overall decrease in speed) was used in all gradient categories, with three direct observations. Firstly, the variability of speed was higher in LEV, and, unlike in UH and DH, speed loss was the greatest in the first half of the event. These observations are characteristic of a subset of the three main types of pacing referred to as parabolic or mixed pacing, with a positive pacing strategy during

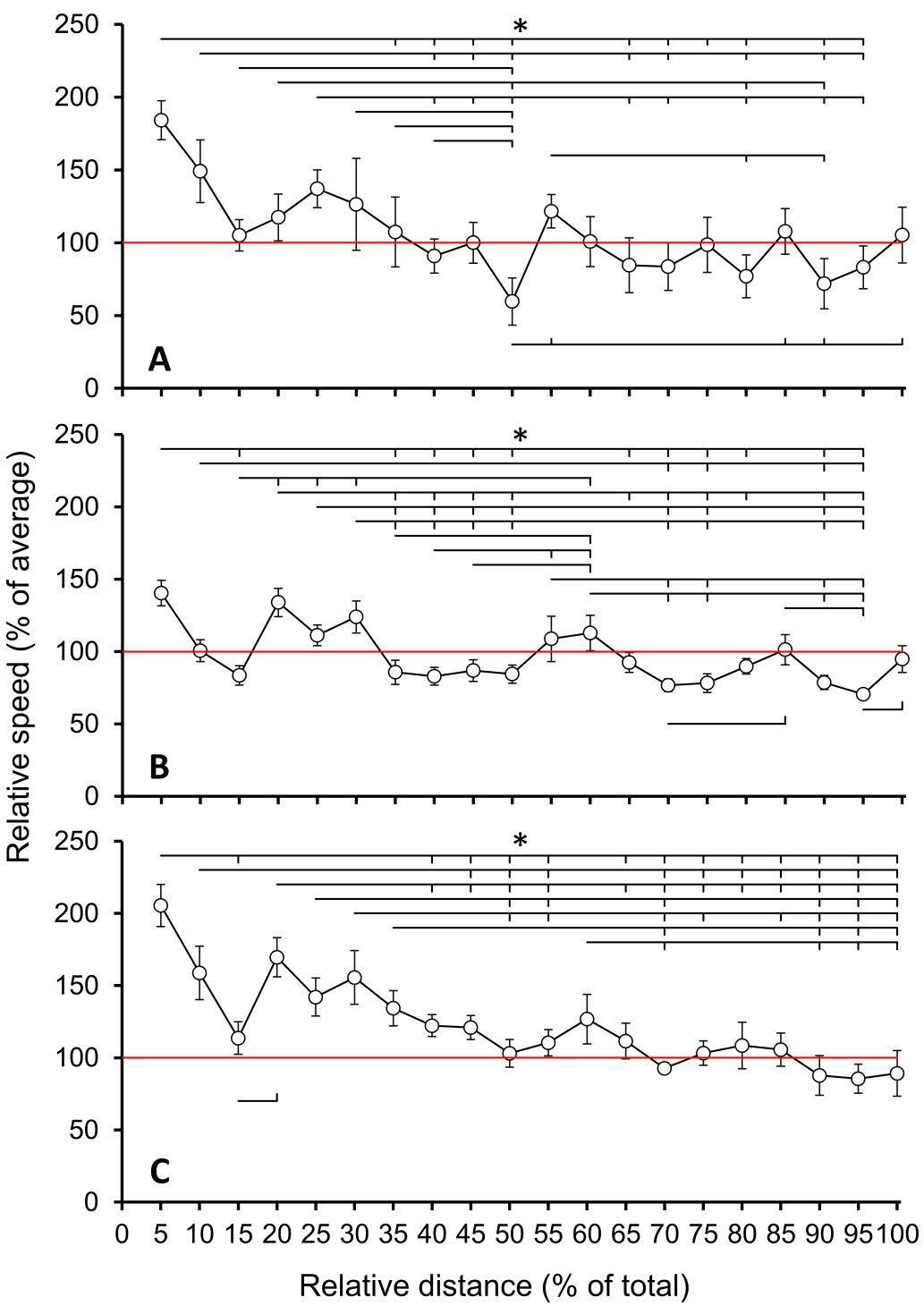

**Figure 2  Dynamics of speed.** Group level (A), uphill (B), and downhill (C) speed relative to average, respectively. Symbols * and brackets are used to denote and locate significant differences ($p < 0.05$).

**Table 1  Correlation between race performance and pacing characteristics.**

| | All participants | | 103-km | | 173-km | |
|---|---|---|---|---|---|---|
| | $r$ | $P$ | $r$ | $p$ | $r$ | $p$ |
| **Speed variability** | −0.23 | 0.42 | −0.24 | 0.70 | −0.25 | 0.48 |
| **Speed loss** | −0.24 | 0.39 | −0.43 | 0.47 | 0.00 | 1.00 |
| **Total time stopped** | −0.35 | 0.21 | −0.15 | 0.81 | −0.45 | 0.20 |

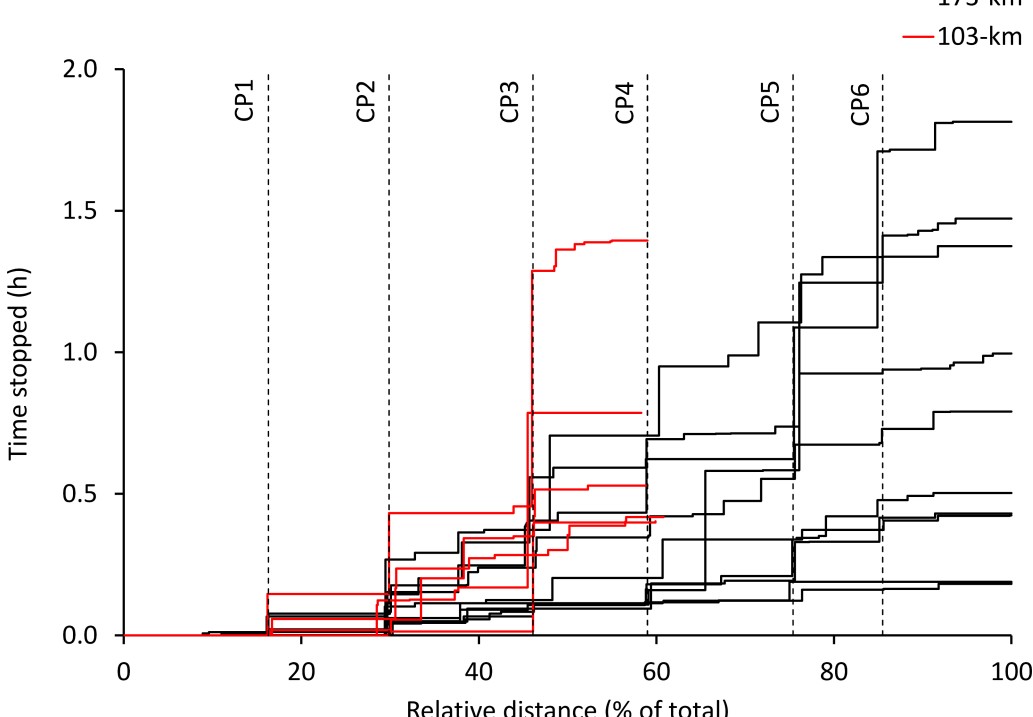

**Figure 3  Total time stopped.** Time stopped as a function of relative distance in all study participants, using the cumulated durations associated with speed $<1$ km h$^{-1}$.

the first half of the event, and more evenly pace for the remainder of the event. Secondly, a speed reserve (increase in the last stage of an event) was measured in LEV and UH. Thirdly, the decrease in speed was greater in DH and LEV compared to UH, and the decrease in relative speed continued until the last section in DH only. Together, these findings clearly indicate that despite slowing down overall, LEV, UH and DH were paced differentially. A secondary finding of this study was that neither the characteristics of relative speed (no significant differences in normality, variances or means), nor the characteristics of pacing (no differences in speed loss, variability of speed or total time stopped) were different in the 103-km and 173-km distances. This allowed us to compare the two groups and increase the relevance of our findings, but this finding also introduces the possibility that the additional ∼70 km did not significantly alter pacing in an ultramarathon performed in hilly terrain.

A recent study from our group also reported parabolic pacing during a 105 km mountain UM, characterised by a decrease of speed at all gradient categories in the later stages of the event (70–90% of total event duration), and by a final increase in speed in the last 10% of the event (*Kerhervé, Millet & Solomon, 2015*). The presence of an increase in speed in the last 10% section of the event was discussed as an indicator of conservative pacing strategies in anticipation of upcoming topographic difficulties, and the use of a speed reserve (*Millet, 2011*) when the last topographic difficulty was passed. In the current study, both the longer distance and smaller elevation gain and loss could potentially explain the absence of a speed reserve in DH. Other research have noted the determinant role of elevation gain and loss in the type and magnitude of fatigue on neuromuscular (*Martin et al., 2010*; *Millet et al., 2011b*) and biomechanical alterations (*Millet et al., 2011a*; *Morin et al., 2011b*). Alternatively, this finding could also potentially highlight the selective effects of fatigue as a function of gradient category, specifically to limit the effect of the transmission of force or vibrations to the musculo-skeletal systems, as previously indicated by the alteration of running economy in LEV and DH gradient categories, but not in UH (*Morin et al., 2011b*; *Vernillo et al., 2015*).

The absence of significant correlations between overall performance and indicators of speed variability, speed loss and total time stopped are novel. While speed loss and total time stopped were found to be significantly correlated with performance in a mountain UM (*Kerhervé, Millet & Solomon, 2015*), speed loss and speed variability were found to be significantly correlated with performance obtained in marathon (*Ely et al., 2008*; *Haney & Mercer, 2011*) and UM (*Hoffman, 2014*; *Lambert et al., 2004*) running. Therefore, additional research is required to determine if specific pacing characteristics are useful predictors of performance in long UM, and whether other variables currently not measured could be better predictors of pacing, including but variables of gait kinematics (stride frequency, stride length, ground contact and aerial times) or gait biomechanics (leg and vertical stiffness), which are known to be influenced by fatigue and gradient (*Morin, Samozino & Millet, 2011a*; *Morin et al., 2011b*; *Vernillo et al., 2016*; *Vernillo et al., 2015*; *Vernillo et al., 2014*). Additionally, it has been proposed that the correlation between the level of performance and a more even pace is due to learning (*Foster et al., 1994*; *Green et al., 2010*), and that the previous practice of a specific distance produces more even pacing (*Ansley et al., 2004*; *Green et al., 2010*). However, ultra-endurance events require longer recovery periods than shorter events and hence, the opportunities to practice a specific distance are relatively less than for shorter events, which could partly explain both the relative lack of data on longer events (*Abbiss & Laursen, 2008*), and the lack of a significant correlation of these variables in the current study.

There were three direct limitations to this study. First, the findings related to the variations of speed within sections of relative distance used binary results (different or not), but do not provide an estimation of the magnitude of differences. A simple level of analysis is not only acceptable, but also warranted, for the type of data used in this study. Future studies are required to describe more accurately the direction and magnitude of changes. Second, the inclusion of participants from both the 103-km and 173-km distances could have implications for the results, due to any differences in *a priori* pacing

strategies across the two distances, or information relative to the decision to stop for the 103-km group. For example, the five participants who had entered the 173-km event and stopped at the 103-km distance might have done so due to an inappropriate pacing strategy, or to a deliberate strategy. Irrespective, we reported the outcomes of pacing, which incorporate both inappropriate and deliberate strategies, and provided evidence that the general patterns of pacing did not differ across groups. Third, there are potential carry-over effects at transitions between types of gradients during running (*Townshend, Worringham & Stewart, 2009*). However, the spatial and temporal resolutions permitted by non-differential GPS do not currently allow sufficient accuracy for this level of analysis during UM events, and future studies using alternative methodologies are necessary to investigate these effects.

## CONCLUSION

In conclusion, in this study we determined the dynamics and characteristics of speed of UM runners in an actual event, which provides a basis for future studies of ultra-long duration exercise. While the speed of all participants decreased as a function of distance over the entire event in all gradient categories, pacing was not comparable in those categories. Finally, overall performance was not correlated to expected predictors of overall running performance (variability of speed, speed loss), or to the total time stopped. Future studies are required to study the dynamics of speed during multiple formats of UM, to determine the effects of *a priori* pacing strategies, distance, and elevation gain and loss, on pacing. As has been done in other ultra-endurance disciplines (*Herbst et al., 2011*), future studies are also warranted to investigate the importance of variables related to participant experience (number of years of practice, number of starts at a certain distance) in order to further characterise pacing and performance in UM events.

## ACKNOWLEDGEMENTS

The authors would like to thank Dr. Andrew Townshend for his valuable comments in the conception of the study.

### Funding

There was no specific research funding for this project. General university (USC, QUT) research funds and scholarships supported the project and two authors (HK: USC and Queensland Education and Training; TC-H: Australian Postgraduate Award). The funders had no role in study design, data collection and analysis, decision to publish, or preparation of the manuscript.

### Grant Disclosures

The following grant information was disclosed by the authors:
USC, QUT research funds.

Australian Postgraduate Award: TC-H.
USC and Queensland Education and Training: HK.

## Competing Interests

The authors declare there are no competing interests.

## Author Contributions

- Hugo A. Kerhervé conceived and designed the experiments, analyzed the data, wrote the paper, prepared figures and/or tables, reviewed drafts of the paper.
- Tom Cole-Hunter performed the experiments, wrote the paper, prepared figures and/or tables, reviewed drafts of the paper.
- Aaron N. Wiegand analyzed the data, wrote the paper, prepared figures and/or tables, reviewed drafts of the paper.
- Colin Solomon conceived and designed the experiments, performed the experiments, analyzed the data, wrote the paper, prepared figures and/or tables, reviewed drafts of the paper.

## Human Ethics

The following information was supplied relating to ethical approvals (i.e., approving body and any reference numbers):

Ethics committee of the Queensland University of Technology, project 0900001233.

## Data Availability

Datasets, R code and results are available at Figshare:

Kerherve H, Cole-Hunter T, Wiegand AN, Solomon S. 2016. Pacing during an ultramarathon running event in hilly terrain. Figshare: https://dx.doi.org/10.6084/m9.figshare.3369790.v1.

## Supplemental Information

Supplemental information for this article can be found online at http://dx.doi.org/10.7717/peerj.2591#supplemental-information.

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
