# Peer review of "Pacing during an ultramarathon running event in hilly terrain"

_PeerJ, doi:10.7717/peerj.2591_

## Round 0.1 · original submission · Major Revisions

· Academic Editor

Major Revisions

This is an interesting article in an area that seems to be under-researched. The reviewers have done a very good job to make recommendations for improvement. I think that in future studies it would also be very interesting to study the evolution of step length and step frequency.

Reviewer 1 ·

Basic reporting

The manuscript could benefit from an editorial trim in some sections. Other than that it conforms to the guidelines.

Experimental design

The authors present an observational paper describing pacing strategies of ultra-mountain endurance athletes over a race distance of 173km (n=10) and 103km (n=5). Time and 3D-position data were recorded using GPS devices (0.2Hz) throughout the duration of the run. Data was distance-normalized (103km = 60%), filtered, centralized (group average speed) and divided into 5% bins. For each bin, relative uphill (>2.5%), downhill (<-2.5%) and level speed (-2.5<LEV<2.5%) were calculated. Over the race, speed decreased in all inclinations, albeit at different rates. A speed reserve was observed in the last bin for some inclinations. The level inclination demonstrated the highest speed variability and also the earliest switch from positive to even pacing. The most significant findings are described as a lack of negative correlation between predictors of performance (time stopped, speed loss, and speed variability) performance (time to finish).
The field of UM marathons is unique in offering a window on fatigue processes in grand magnitude race events. The acquisition of reliable continuous position and (psycho)physiological data is challenging and there is an enormous amount of confounders to be taken into account. The authors have done a labor intensive job collecting and cleaning GPS data from so many participants over such a long distance.
The contribution to the current base of knowledge lies mainly in the addition of a longer data set compared to a previous paper from the same group which assessed more outcome variables. The observations are subtly different between the two papers, begging the question if pacing is not more a function of the individual race profile and terrain than in any way indicative of a fatigue model. Also the sensitivity and specificity of pacing as a marker of fatigue may be severely compromised in these types of outdoor sports in which the environmental constraints (temperature, weather, trail condition, exposition, importance, etc.) may prove strong determinants of speed fluctuations. Placing the reported observations into the context of current models of fatigue and interpreting them within this framework can significantly strengthen the paper and make it more interesting. The authors are experts in this area having published multiple papers on similar topics.
The reported measures relate only to the dynamics of velocity management. There are no assessments of other fatigue indices on neither a physiological nor subjective level. This is disappointing as it would provide more relevance to the reported outcomes. I strongly agree with the authors’ final conclusion that “future studies are also warranted to investigate the importance of variables related to participant experience (number of years of practice, number of starts at a certain distance) in order to further characterise pacing and performance in UM events.”. More velocity data from different types of UM is needed to generate and validate a comprehensive framework for fatigue development in UM events and therefore this is a useful addition.

Validity of the findings

No Comments

Additional comments

Major comments
Why would you expect pacing to vary as a function of inclination? While it seems obvious that different inclines will result in different initial speeds, I believe a paragraph is warranted describing the framework that leads you to believe that speed dynamics evolve differentially in different inclinations.
How have you taken into account carry-over effects from earlier inclinations? Should this be mentioned in the limitations if not taken into account?
(Townshend, Andrew D. and Worringham, Charles J. and Stewart, Ian (2009) Spontaneous pacing during overground hill running. Medicine and Science in Sports and Exercise, 42(1). pp. 160-169.)
In the discussion once again I feel that there is not enough discussion (or even speculation) concerning the latent factors driving differential pacing on different inclines.
One of the novelties in the reported results lies in that all participants started under the same environmental conditions with the same aim, however some failed to complete that aim. More detail on the groups could make this into a much more interesting paper.
Do you believe that speed variability in the LEV condition is not driven by minor (indistinguishable) changes in gradient? As base velocity is higher, it might be expected that a given gradient change may have greater impact as compared to uphill for example.

Minor comments:
Line 48/49: Stating that these ultra-long type of events enable the study of fatigue is I believe slightly too general. The fatigue that is studied in these events is specific to the nature of the event and also to the rather specialized population.
Line 62: Please make it clearer in the first sentence that these findings are from a different study.
Line 66-68: Are the populations comparable in terms of expertise, performance and motivation between these two studies? Are the course characteristics comparable (exposition, temperature, environmental constraints (altitude, snow, etc.)?
Line 76: The causality of this seems inverted: Please turn around to state that “the aim is to assess pacing, therefore we will use GPS” and not “because we have GPS, we will investigate pacing”.
Line 85: Maybe mention the race name so people can assess what kind of terrain it was conducted in?
Line 118: If you replaced all the missing values with 0, that would impact your means and your variability.
Line 121: “This procedure limited” might be more elegant.
Line 122-24: Maybe “zero-speed values associated with checkpoint locations were exempt from this treatment.”. I’m guessing you excluded these sections before smoothing?
Line 154-156: I find this very confusing. I’m guessing that it to some point reflects the “sparseness” of some datasets? In some cases it would be interesting to know how many points of each type are in each bin. It seems that the statistical analysis may be biased in those bins with a large skew toward any inclination.
Line 179: How about climate conditions? It would be good if you could give the temperature bounds and precipitation as temperature will impact pacing and precipitation will make the terrain more difficult to navigate.
Line 180: I think some demographics of the subjects should be presented – at minimum their level of experience with trail running and their age. Also whether they had prior knowledge of the course. I’m guessing they mere highly motivated, however is there any measure of their risk-taking behavior? This might be interesting for interpreting descent speed on technical terrain.
Line 181: I’m assuming all participants did aim to complete the 173km? Otherwise, seeing that we are taking into account anticipatory pacing and feed forward regulation this would be expected to have had an effect...
Line 202: In the study you are not just reporting, but also collecting data… Also line 203 please revise wording
Line 205: Overall decrease in speed
Line 207: “main component was greatest” – doubled expression?
Line 214: What are the further implications of the groups demonstrating no difference in pacing? Just a few ideas: If the 103km group would be designated as “lower performers” and the 173km group as “higher performers”, would this not indicate that pacing is completely insensitive to performance level? Why did the runners stop at 103km? Tactical decision? Were they fully fatigued? Were they equally fatigued as the 173km group? Did they pay a higher cardio-vascular price to adhere to the pacing profile? How about their absolute speeds – were these different? …
Line 219: No “a”
Line 227: In figure 2, there seems to be a significant speed increase in the last 10% of LEV and the last 5% of UH - Would this not constitute the speed reserve? Also, from the race profile in figure 1, the last DH segment looks steeper than the other ones. Do you not believe that the inclination and technicality of the last segment may have impacted the speed more than any pacing strategy?
Line 254: I don’t see how this paper provides a “basis for future studies of ultra-long duration exercise”. To my comprehension, the basis/framework has been provided in other publications and this is an addition of observational data.

Reviewer 2 ·

Basic reporting

This is a quite interesting and well-written article that expands the knowledge on some aspects of a still poorly studied research area. The introduction and background are adequate and figures and tables are complete and informative, although the large number of data represented in the figures makes their reading not always immediate. However, there are some points I would like to discuss with the Authors:

Paragraph 1, lines 34-36: Although the statement is correct, it should be noted how the principal reason of studying pacing strategy relates to the understanding of physiological and regulatory processes in function of the optimization of exercise performance, which does not only depend on bioenergetics. I would suggest to the Authors to rewrite this phrase, also in consideration that the article does not focus on the metabolic aspects of pacing.

Paragraph 1, line 40: I do not have access to the full article (Firth 1998).

Paragraph 1, line 42: Please note that in the reported study (Tucker et al. 2004) the Authors measured pacing strategies only on 800-, 5000- and 1000-meter distances. Moreover, I am not sure that defining the running dynamics observed in the 5000- and 1000-meters by Tucker and colleagues as “negative pacing” is correct. According to the classification given by Abiss & Laursen (2008), the most appropriated adjective for that pacing strategy is “parabolic” or, alternatively, “mixed” (according to the classification given by the Authors of the present study).

Paragraph 2, line 52: In line with the previous comment, the presence of a speed reserve detected by Kerhervé et al. (2015) suggests a parabolic/mixed rather than a positive pacing strategy.

Paragraph 2, line 55: Note that in the studies of Davies & Thompson (1979) and Millet et al. (2011) pacing has not been measured. Moreover, despite gradual increases in heart rate and VO2 have been observed in the study of Davies & Thompson (1986), I would use caution in interpreting those cardiovascular changes as a pacing strategy, since the treadmill speed was maintained constant for the whole duration of the test.

Paragraph 2, line 59: Angus & Waterhouse (2011) did not report differences in pacing variation between the three cohorts of running speeds assessed on ultramarathon distance.

Paragraph 9, lines 126-131: This part is copied-and-paste from the Methods section of the article of Kerhervé et al. (2015). This might be considered self-plagiarism and therefore be in contrast with the of the Journal policy on Publication Ethics. Please rewrite this part.

Raw data: Although the raw data have been made available as a part of Supplemental files and on the Figshare data repository, their reading results unclear and confusing since they have not been stacked and labelled in a clear manner.

Experimental design

The aims of the study are clear and the information about how GPS method has been used to assess pacing characteristics are described sufficiently in details. However:

Paragraph 7, line 107: A brief explanation on how to use the Vincenty formulae to measure point-to-point distances might facilitate the reading comprehension.

Paragraph 10, lines 150-153: What is the rationale and/or the scientific evidence used to determine these gradient ranges (LEV: -2.5 to 2.5%; UH: 2.5 to 100% and DH: -100 to -2.5%)?

Validity of the findings

Paragraph 14, lines 176-179/ Paragraph 22, lines 247-250: Please specify (if the Authors have access to those information) the reason of the pre-termination of the race (exhaustion, adverse weather conditions, etc) as it may have important implications on the interpretation of the results.

Paragraph 16, lines 205-213: There are no information about the characteristics of participants (training level/background) and about environmental conditions (temperature, wind, rainfall, etc.) occurred during the competition. Since these variables have been demonstrated to affect pacing strategies (Abiss & Laursen 2008; Hoffman 2014), their omission may have affected the power of data interpretation. Moreover, by uniting all the data according with their gradient range (LEV: -2.5 to 2.5%; UH: 2.5 to 100% and DH: -100 to -2.5%), how the Authors can exclude that the observed pacing variation at a given gradient has not been affected by the other interludes occurred within a given gradient range?

Paragraph 16, lines 213-216: Although using the average running speed as index of running performance permits to compare the two different groups (173-km and 103-km participants), have the Authors also tried to correlate the pacing characteristics with the finishing time in the two groups separately before concluding that there is no correlation between these characteristics and endurance performance?

Abstract, lines 25-28: This sentence is unclear: how positive pacing could have characterized all gradients if a speed reserve has been observed in LEV and UH?

Abstract, lines 28-30: This conclusion does not seem valid since significant correlations between overall performance and pacing characteristics have been previously observed in 161-km mountain ultramarathons (Hoffman 2014).

Additional comments

As I previously stated, this article is certainly interesting and it provides insight into some not well studied aspects of exercise science through a systematic analysis of pacing in a trail ultramarathon running event. However, there are some parts that definitely need to be corrected and/or rewritten in order to make it publishable.

---

## Round 0.2 · Minor Revisions

· Academic Editor

Minor Revisions

Please take into account the suggestions of reviewer 2, which should in principle be doable but will need some work, and have a look at the recommendation I had given in my first reply.

Reviewer 1 ·

Basic reporting

Clarity of the manuscript has increased following the revision. The main content has not been modulated extensively, nor are the conclusions in any way different. They continue to conform to the structure of the underlying data. The rewriting of the introduction has strengthened this section.

Experimental design

No Comments

Validity of the findings

No Comments

Additional comments

The authors have addressed all the points remonstrated in the first review and provided adaptations or replies. This has to a certain extent improved the manuscript. However, the lack of context for the reported observations increases the difficulty of interpretation and decreases the usefulness of this publication. Within this framework I remain of my former opinion that this type of data is important for the field, however it needs to be collected and reported in a conscientious manner with as much surrogate and corroborate data as possible, especially those data that are known to affect pacing strategy.

Reviewer 2 ·

Basic reporting

The authors have made most of the recommended corrections, other than having clarified some critical parts. However, a couple of points still require minor revisions.

Although substantial amendments have been made following the suggestions of the two Reviewers, this procedure has partially disrupted the flow of the article. Specifically, from the Paragraph 2 the article starts focusing on ultramarathon (UM) distances without a declared motivation (perhaps the increasing popularity of the events associated with the paucity of studies available on the topic?). Importantly, the aims of the study (Paragraph 4, lines 72-76) are not solidly supported by the previous paragraphs since the part about the effects of fatigue as a function of gradient category has been described only in the following phrase (Paragraph 4, lines 76-80). In this regard, I would suggest to put that part at the end of the Paragraph 3, before the enunciation of the study purposes.

Despite the article from Angus & Waterhouse (2011) (Paragraph 2, line 59 of the previous version of the manuscript) has been properly removed from the introduction, it is still present in the discussion (Paragraph 21, line 268-269 of the current version of the manuscript). Since in that article the authors did not observe major differences in pacing variation between the three cohorts of running speeds assessed on UM distance, I would recommend to remove it or replace it with Hofmann (2014).

Experimental design

No Comments

Validity of the findings

No Comments

---

## Author Rebuttal · Round 0.2

Hugo Kerherve

hkerherv@usc.edu.au

August 5th, 2016

Dear Editor,

We would like to thank the editor and the reviewers for the time and effort taken to review and make the extremely valuable comments for the improvement of this article.

We have carefully reviewed all the comments and have completed a point-by-point response, where applicable, which is provided below.

We have made substantial amendments following the comments from the reviewers. We believe these have significantly improved the quality of the article, and hope these meet with the approval of the editor and reviewers.

Sincerely Yours,

Hugo Kerhervé

# Reviewer 1

## Basic reporting

The manuscript could benefit from an editorial trim in some sections. Other than that it conforms to the guidelines.

## Experimental design

The authors present an observational paper describing pacing strategies of ultra-mountain endurance athletes over a race distance of 173km (n=10) and 103km (n=5). Time and 3D-position data were recorded using GPS devices (0.2Hz) throughout the duration of the run. Data was distance-normalized (103km = 60%), filtered, centralized (group average speed) and divided into 5% bins. For each bin, relative uphill (>2.5%), downhill (<-2.5%) and level speed (-2.5<LEV<2.5%) were calculated. Over the race, speed decreased in all inclinations, albeit at different rates. A speed reserve was observed in the last bin for some inclinations. The level inclination demonstrated the highest speed variability and also the earliest switch from positive to even pacing. The most significant findings are described as a lack of negative correlation between predictors of performance (time stopped, speed loss, and speed variability) performance (time to finish).

The field of UM marathons is unique in offering a window on fatigue processes in grand magnitude race events. The acquisition of reliable continuous position and (psycho)physiological data is challenging and there is an enormous amount of confounders to be taken into account. The authors have done a labor intensive job collecting and cleaning GPS data from so many participants over such a long distance.

The contribution to the current base of knowledge lies mainly in the addition of a longer data set compared to a previous paper from the same group which assessed more outcome variables. The observations are subtly different between the two papers, begging the question if pacing is not more a function of the individual race profile and terrain than in any way indicative of a fatigue model. Also the sensitivity and specificity of pacing as a marker of fatigue may be severely compromised in these types of outdoor sports in which the environmental constraints (temperature, weather, trail condition, exposition, importance, etc.) may prove strong determinants of speed fluctuations. Placing the reported observations into the context of current models of fatigue and interpreting them within this framework can significantly strengthen the paper and make it more interesting. The authors are experts in this area having published multiple papers on similar topics.

The reported measures relate only to the dynamics of velocity management. There are no assessments of other fatigue indices on neither a physiological nor subjective level. This is disappointing as it would provide more relevance to the reported outcomes. I strongly agree with the authors' final conclusion that "future studies are also warranted to investigate the importance of variables related to participant experience (number of years of practice, number of starts at a certain distance) in order to further characterise pacing and performance in UM events.". More velocity data from different types of UM is needed to generate and validate a comprehensive framework for fatigue development in UM events and therefore this is a useful addition.

## Validity of the findings

No Comments

# Comments for the author

## Major comments

Reviewer's comment: Why would you expect pacing to vary as a function of inclination? While it seems obvious that different inclines will result in different initial speeds, I believe a paragraph is warranted describing the framework that leads you to believe that speed dynamics evolve differentially in different inclinations.

We have added a section in paragraph #4 following the reviewer's comment.

> **Paragraph #4.** (…)The literature specific to UM and ultra-endurance exercise has highlighted the protective nature of fatigue in situations where the physical integrity of participants could be threatened, such as greater neuromuscular (Millet et al. 2011b) or biomechanical alterations (Morin et al. 2011b) in events with large elevation gain and loss compared to UM events on level ground (Martin et al. 2010; Morin et al. 2011a). Therefore, speed was expected to decrease in all participants and at all gradients as a function of the distance and duration of the event.

How have you taken into account carry-over effects from earlier inclinations? Should this be mentioned in the limitations if not taken into account? (Townshend, Andrew D. and Worringham, Charles J. and Stewart, Ian (2009) Spontaneous pacing during overground hill running. Medicine and Science in Sports and Exercise, 42(1). pp. 160-169.)

We thank the reviewer for this valuable comment. The carry-over effect is one of the most interesting phenomena, which deserves independent emphasis in future research in UM running. We have attempted to generate such findings in this study. However, the temporal resolution permitted with the current dataset does not allow us to discriminate with sufficient accuracy the discrete locations of topographic events (LEV before/after UH or DH, for example).

> **Paragraph #22.** There were three direct limitations to this study. (…) Third, there are potential carry-over effects at transitions between types of gradients during running (Townshend et al. 2009). However, the spatial and temporal resolutions permitted by non-differential GPS do not currently allow sufficient accuracy for this level of analysis during UM events, and future studies using alternative methodologies are necessary to investigate these effects.

In the discussion once again I feel that there is not enough discussion (or even speculation) concerning the latent factors driving differential pacing on different inclines.

We thank the reviewer for this valuable comment. As the reviewer noted, this is a descriptive study. As such, we believe we should refrain from speculation. We believe the changes provided in the introduction regarding the main factors governing the selective variation of speed at different gradients (see first reviewer's comment) provide the primary basis for discussion. We have therefore included these statements in the discussion.

> **Paragraph #20.** (…) Other research have noted the determinant role of elevation gain and loss in the type and magnitude of fatigue on neuromuscular (Martin et al. 2010; Millet et al. 2011b) and biomechanical alterations (Millet et al. 2011a; Morin et al. 2011b). Alternatively, this finding could also potentially highlight the selective effects of fatigue as a function of gradient category, as previously indicated by the alteration of running economy in LEV and DH gradient categories, where speed is higher than during uphill locomotion (Morin et al. 2011b; Vernillo et al. 2015).

One of the novelties in the reported results lies in that all participants started under the same environmental conditions with the same aim, however some failed to complete that aim. More detail on the groups could make this into a much more interesting paper.

We agree with the reviewer. However, we do not have such information. We do express this as a limitation of the current study, and recommend for future studies to complement such measures of pacing, using a priori pacing strategies, as well as other measures of perceived exertion, pain, or mood states.

**Paragraph #22.** (…) Second, the inclusion of participants from both the 103-km and 173-km distances could have implications for the results, due to any differences in *a priori* pacing strategies across the two distances, or information relative to the decision to stop for the 103-km group. For example, the five participants who had entered the 173-km event and stopped at the 103-km distance might have done so due to an inappropriate pacing strategy, or to a deliberate strategy. Irrespective, we reported the outcomes of pacing, which incorporate both inappropriate and deliberate strategies, and provided evidence that the general patterns of pacing did not differ across groups. (…)

**Paragraph #23.** (…) Future studies are required to study the dynamics of speed during multiple formats of UM, to determine the effects of *a priori* pacing strategies, distance, and elevation gain and loss, on pacing.

**Do you believe that speed variability in the LEV condition is not driven by minor (indistinguishable) changes in gradient? As base velocity is higher, it might be expected that a given gradient change may have greater impact as compared to uphill for example.**

We thank the reviewer for this excellent comment. We believe that speed variability is not driven by minor changes in gradients, when considering the constraints of the current study. The greater inertia associated with the greater speed on LEV compared to UH or DH, would be only marginally affected by the relatively narrow range of gradients used in this study (±2.5% is barely perceptible),

The accuracy of the system of measure and the necessary data treatment (filtering + smoothing + section averaging) both contribute to ensure that these changes are indeed, indistinguishable in the current state of non-differential GPS technology.

There is potential that speed variability in LEV may be affected by minor changes in gradient more than in UH, for two reasons. Firstly, a small topographic obstacle (short and steep UH or DH) would have a minor effect on speed selection. Secondly, a lower variability of speed is expected for the UH gradients, characterised by both a lower mean speed and a lower loss of speed, compared to the LEV and DH gradients.

## Minor comments:

**Line 48/49: Stating that these ultra-long type of events enable the study of fatigue is I believe slightly too general. The fatigue that is studied in these events is specific to the nature of the event and also to the rather specialized population.**

We have revised Paragraphs 1 and 2 following Reviewer #2's comments. This statement has been removed.

**Line 62: Please make it clearer in the first sentence that these findings are from a different study.**

We have revised the section according to the reviewer's comment.

**Paragraph #3.** In contrast to these findings, we measured in a previous study on trained UM runners a higher magnitude of speed loss in faster compared to slower runners, no significant relationship between the variability of speed and performance level, and a novel significant negative relationship between the total time stopped and performance level, in a long mountain UM (Kerhervé et al. 2015)

**Line 66-68:** Are the populations comparable in terms of expertise, performance and motivation between these two studies? Are the course characteristics comparable (exposition, temperature, environmental constraints (altitude, snow, etc.)?

The populations were comparable in expertise and motivation, and represented a wide range of performance levels in both events (including a representative range of finish times).

The course characteristics are not comparable. The mountainous terrain of the UM in Kerhervé et al. (2015) actually prevented us from using better temporal resolution, which is improved in the current study by a factor of two.

**Line 76:** The causality of this seems inverted: Please turn around to state that "the aim is to assess pacing, therefore we will use GPS" and not "because we have GPS, we will investigate pacing".

Agreed. In the interest of concision and clarity, we have removed this information as it was informative, but not essential to the presentation of the study.

**Line 85:** Maybe mention the race name so people can assess what kind of terrain it was conducted in?

The section has been revised according to the reviewer's suggestion.

> **Paragraph #5.** This study was approved by the university research ethics committee (Queensland University of Technology, project 0900001233). The study participants were recruited using advertisements on a specialised forum and researchers networks, from individuals already registered to compete in the Great North Walk 100s (NSW, Australia), a long (~173 km) and hilly UM running event including 6 checkpoints and a total elevation gain and loss of approximately 3,000 m.

**Line 118:** If you replaced all the missing values with 0, that would impact your means and your variability.

This is correct, and is the reason why we chose this approach for the first step of data treatment procedures. This combination of filtering and smoothing actually permitted us to decrease the effect of both signal drift and signal jamming (artifacts artificially increasing distance and speed due to erroneous values), which are even further increased with a nearest-neighbour approach, a common strategy to replace missing values consisting of calculating the arithmetic mean between two acceptable values. The effect of the zero values is decreased by the subsequent data smoothing procedure.

Overall, this two-step procedure permitted us to decrease the effect of erroneous values on distance and speed, while being sensitive to actual periods of rest (zero speed).

**Line 121:** "This procedure limited" might be more elegant.

Wording has been modified following the reviewer's suggestion.

> **Paragraph #8.** (…) This procedure limited the decrease of the effect of signal drift and jamming (higher distance and speed due to erroneous values).

**Line 122-24:** Maybe "zero-speed values associated with checkpoint locations were exempt from this treatment.". I'm guessing you excluded these sections before smoothing?

We did not exclude sections of zero-speed values: this section means, there were sections of zero speed values even after applying the filtering and smoothing procedures (zero speed values are not observed using either raw or nearest-neighbour approach).

Additionally, these zero-speed values occurred at expected locations of checkpoints in the race. We modified the wording to the plural form ("These procedures were…") to alleviate the confusion.

> **Paragraph #8.** (…) These procedures were sensitive to periods of zero speed values, which corresponded to the location, via expected relative distances, of checkpoints in the race.

Line 154-156: I find this very confusing. I'm guessing that it to some point reflects the "sparseness" of some datasets? In some cases it would be interesting to know how many points of each type are in each bin. It seems that the statistical analysis may be biased in those bins with a large skew toward any inclination.

We have considered this matter, and to the contrary, we think this information actually reinforces the decision to use 5% instead of 10% section averages, since the smallest amount of data amounts to 2.76 km in one gradient category in a section (approximately 8.5 km) including a large downhill component. Additionally, we think this information is beneficial to justify the use of inferential statistics in the sense that there were always sufficient data points in each bins.

Since the statistical analysis uses the average of each bin, and that each gradient was treated independently of each other, there should be no skewness towards larger bins. We have acknowledged both the broad definitions of each inclinations as well as the type of inferential statistics used as limitations.

In the interest of concision and clarity, we have provided the mean and standard deviation of the number of data points per bin for each inclinations.

Line 179: How about climate conditions? It would be good if you could give the temperature bounds and precipitation as temperature will impact pacing and precipitation will make the terrain more difficult to navigate.

We have added information about the weather conditions (daily total rainfall, minimum and maximum temperatures) retrieved from the database of the Australian Bureau of Meteorology (http://www.bom.gov.au/climate/data/stations/) at the nearest weather stations.

Please also refer to similar comment from Reviewer #2.

> **Paragraph #14.** (…) Weather conditions on race day were dry (no precipitations), the temperatures ranged 14.9-22.8 ℃ at the start point, 10.2-29.1 ℃ at the 103-km checkpoint, and 11.4-26.8 ℃ at the finish line, respectively.

Line 180: I think some demographics of the subjects should be presented – at minimum their level of experience with trail running and their age. Also whether they had prior knowledge of the course. I'm guessing they mere highly motivated, however is there any measure of their risk-taking behavior? This might be interesting for interpreting descent speed on technical terrain.

Unfortunately, we do not possess such information. However, previous systematic descriptions of pacing in UM (Lambert 2004, Angus 2011) did not either. Since the current study remains at a descriptive level for pacing and performance outcomes, and does not consider other parameters, we believe this does not hinder the applicability of findings to other similar events.

Line 181: I'm assuming all participants did aim to complete the 173km? Otherwise, seeing that we are taking into account anticipatory pacing and feed forward regulation this would be expected to have had an effect…

All participants were registered for, and intended to run the 173 km distance.

### Line 202: In the study you are not just reporting, but also collecting data… Also line 203 please revise wording

Agreed. The wording has been modified following the reviewer's suggestion.

> **Paragraph #18.** In this study, we collected and reported the longest systematic description of pacing of runners in a long, hilly UM running event, using a method that created no disturbances to normal running event situations.

### Line 205: Overall decrease in speed

Agreed. The wording has been modified following the reviewer's suggestion.

> **Paragraph #19.** The primary finding of this study was that positive pacing (overall decrease in speed) was used in all gradient categories, with three direct observations. (…)

### Line 207: "main component was greatest" – doubled expression?

Agreed. Wording has been modified using "speed loss" instead of "the main component of speed loss".

> **Paragraph #19.** (…) Firstly, the variability of speed was higher in LEV, and, unlike in UH and DH, the main component of speed loss was the greatest in the first half of the event.

### Line 214: What are the further implications of the groups demonstrating no difference in pacing? Just a few ideas: If the 103km group would be designated as "lower performers" and the 173km group as "higher performers", would this not indicate that pacing is completely insensitive to performance level? Why did the runners stop at 103km? Tactical decision? Were they fully fatigued? Were they equally fatigued as the 173km group? Did they pay a higher cardio-vascular price to adhere to the pacing profile? How about their absolute speeds – were these different? …

The reviewer is correct, and these questions warrant future studies to measure pacing, pacing strategies, and subjective perceptions in participants finishing and not finishing UM events. However, we believe the current data does not allow us to provide this level of analysis. Additionally, event organisations offering the opportunity to receive a classification for completing part of the event are not common, and therefore labelling participants having not finished the 173-km event as "low performers" would limit the applicability of such an approach to future studies.

### Line 219: No "a"

Agreed. Wording has been modified following the reviewer's suggestion.

> **Paragraph #19.** (…) This allowed us to compare the two groups and increase the relevance of our findings, but this finding also introduces the possibility that the additional ~70 km did not significantly alter pacing in an ultramarathon performed in a hilly terrain.

### Line 227: In figure 2, there seems to be a significant speed increase in the last 10% of LEV and the last 5% of UH - Would this not constitute the speed reserve? Also, from the race profile in figure 1, the last DH segment looks steeper than the other ones. Do you not believe that the inclination and technicality of the last segment may have impacted the speed more than any pacing strategy?

The reviewer is correct. This was a carry-over from a previous version when generating the LaTex file, and the sentence has been modified accordingly.

> **Paragraph #20.** (…) In the current study, both the longer distance and smaller elevation gain and loss could potentially explain the absence of a speed reserve in DH.

Line 254: I don't see how this paper provides a "basis for future studies of ultra-long duration exercise". To my comprehension, the basis/framework has been provided in other publications and this is an addition of observational data.

We respectfully partially disagree with the reviewer, in that the framework was so far predictive (pacing in UM was expected to consist of positive pacing with a speed reserve) but so far not observed in data.

No other research group had so far generated an actual observational study of UM performance in hilly and mountainous terrain, yet many studies speculatively explain their findings in relation to UM performance. The series of studies on pacing in hilly (current study) and mountainous UM (Kerherve et al. 2015) provides a methodological basis for the measurement of pacing, which can now be completed using other variables in order to generate mechanistic and explanatory perspectives of UM performance.

# Reviewer 2

## Basic reporting

This is a quite interesting and well-written article that expands the knowledge on some aspects of a still poorly studied research area. The introduction and background are adequate and figures and tables are complete and informative, although the large number of data represented in the figures makes their reading not always immediate. However, there are some points I would like to discuss with the Authors:

Paragraph 1, lines 34-36: Although the statement is correct, it should be noted how the principal reason of studying pacing strategy relates to the understanding of physiological and regulatory processes in function of the optimization of exercise performance, which does not only depend on bioenergetics. I would suggest to the Authors to rewrite this phrase, also in consideration that the article does not focus on the metabolic aspects of pacing.

We thank the reviewer for this comment, and have amended the sentence following their suggestion.

> **Paragraph #1.** The dynamics of speed during self-paced locomotor exercise, or pacing, are used in recreational, competitive and scientific settings as an indicator of exercise intensity (Abbiss & Laursen 2008), and fatigue (Knicker et al. 2011).

Paragraph 1, line 40: I do not have access to the full article (Firth 1998).

The section has been removed based on the following comment.

Paragraph 1, line 42: Please note that in the reported study (Tucker et al. 2004) the Authors measured pacing strategies only on 800-, 5000- and 1000-meter distances. Moreover, I am not sure that defining the running dynamics observed in the 5000- and 1000-meters by Tucker and colleagues as "negative pacing" is correct. According to the classification given by Abiss & Laursen (2008), the most appropriated adjective for that pacing strategy is "parabolic" or, alternatively, "mixed" (according to the classification given by the Authors of the present study).

Agreed, the definitions of types of pacing was simplistic and therefore not factual. Paragraphs 1 and 2 have been largely amended for more careful and accurate wording.

Please also refer to the following comment for a complete answer to the reviewer's comment.

Paragraph 2, line 52: In line with the previous comment, the presence of a speed reserve detected by Kerhervé et al. (2015) suggests a parabolic/mixed rather than a positive pacing strategy.

Agreed, we have now amended the entire section following the reviewer's comment.

Please also refer to the previous comment for a complete answer to the reviewer's comment.

Paragraph 2, line 55: Note that in the studies of Davies & Thompson (1979) and Millet et al. (2011) pacing has not been measured. Moreover, despite gradual increases in heart rate and VO2 have been observed in the study of Davies & Thompson (1986), I would use caution in interpreting those cardiovascular changes as a pacing strategy, since the treadmill speed was maintained constant for the whole duration of the test.

We thank the reviewer for their accurate comment. We have removed the Davies and Thompson (1979) and Davies and Thompson (1986) references, and have replaced the Millet et al. (2011) with Gimenez et al. (2013) from the same 24 h treadmill study which actually reports pacing.

> **Paragraph #2.** (…) Positive pacing has also been observed in other forms of ultra-endurance exercise, such as a 24 h treadmill run (Gimenez et al. 2013), or during an ultra-endurance triathlon event consisting of ten consecutive Ironman distance triathlons (10 x 3.8 km swimming, 180 km cycling, 42 km running) in 10 days (Herbst et al. 2011).

Paragraph 2, line 59: Angus & Waterhouse (2011) did not report differences in pacing variation between the three cohorts of running speeds assessed on ultramarathon distance.

The reviewer is correct and the reference has been removed.

Paragraph 9, lines 126-131: This part is copied-and-paste from the Methods section of the article of Kerhervé et al. (2015). This might be considered self-plagiarism and therefore be in contrast with the of the Journal policy on Publication Ethics. Please rewrite this part.

We have modified the section following the reviewer's suggestion.

> **Paragraph #9.** We used previously published procedures (Kerhervé et al. 2015) to alleviate the inaccuracies in GPS-based elevation (Townshend et al. 2008).

Raw data: Although the raw data have been made available as a part of Supplemental files and on the Figshare data repository, their reading results unclear and confusing since they have not been stacked and labelled in a clear manner.

We acknowledge the files could be clearer, however, we believe this minor inconvenience is far outweighed having provided the data on a simple format (csv) readable by anyone, along with the actual code used in a free open-source computing package (R). The extensively labelled files developed by the authors in excel format are also too large for sharing purposes, and still require a specific understanding of the data to be easily readable.

## Experimental design

The aims of the study are clear and the information about how GPS method has been used to assess pacing characteristics are described sufficiently in details. However:

Paragraph 7, line 107: A brief explanation on how to use the Vincenty formulae to measure point-to-point distances might facilitate the reading comprehension.

We have amended the section following the reviewer's comment. However, in the interest of readability, we focused more on the justification leading to the choice of this method of measure rather than on the detail of the iterative processes for convergence into the solutions.

> **Paragraph #7.** The distance between points at the surface of a sphere can be calculated using simple spherical trigonometry. However, the earth is not a sphere, but an oblate spheroid akin to an ellipsoid with the following dimensions: equatorial radius $\approx 6{,}378.137$ km, polar radius $\approx 6{,}356.752\,314\,245$ km and flattening $f \approx 1/298.257223563$ (Defense Mapping Agency 1990). The calculation of point to point distances at the surface of an ellipsoid can be improved compared to spherical trigonometry using the inverse Vincenty formulae (Vincenty 1975). The point-to-point distances were obtained from an internet-based utility (GPS Visualizer; www.gpsvisualizer.com) using geographical positions (latitude and longitude). We found the distances obtained were in exact

agreement (r = 1.00, p < 0.001) with our preliminary measures of point-to-point distances using the Vincenty formulae ~~(Vincenty 1975)~~ performed on 10 data sets.

### Paragraph 10, lines 150-153: What is the rationale and/or the scientific evidence used to determine these gradient ranges (LEV: -2.5 to 2.5%; UH: 2.5 to 100% and DH: -100 to -2.5%)?

These gradient categories were defined operationally to be as large as possible in order to ensure that a sufficient amount of data points were used, given the known limitations of the system of measurement and of the automated calculation of elevation.

Future studies using higher spatial and temporal accuracies are required to determine pacing in smaller sub-categories of gradient.

## Validity of the findings

### Paragraph 14, lines 176-179/ Paragraph 22, lines 247-250: Please specify (if the Authors have access to those information) the reason of the pre-termination of the race (exhaustion, adverse weather conditions, etc) as it may have important implications on the interpretation of the results.

Unfortunately, we do not have this type of information. We do express this as a limitation, and recommend for future studies collecting such data. We are confident that weather conditions did not play a significant role.

> **Paragraph #22.** (…) Second, the inclusion of participants from both the 103-km and 173-km distances could have implications for the results, due to any differences in *a priori* pacing strategies across the two distances, or information relative to the decision to stop for the 103-km group. For example, the five participants who had entered the 173-km event and stopped at the 103-km distance might have done so due to an inappropriate pacing strategy, or to a deliberate strategy. Irrespective, we reported the outcomes of pacing, which incorporate both inappropriate and deliberate strategies, and provided evidence that the general patterns of pacing did not differ across groups. (…)

> **Paragraph #23.** (…) Future studies are required to study the dynamics of speed during multiple formats of UM, to determine the effects of *a priori* pacing strategies, distance, and elevation gain and loss, on pacing.

### Paragraph 16, lines 205-213: There are no information about the characteristics of participants (training level/background) and about environmental conditions (temperature, wind, rainfall, etc.) occurred during the competition. Since these variables have been demonstrated to affect pacing strategies (Abiss & Laursen 2008; Hoffman 2014), their omission may have affected the power of data interpretation. Moreover, by uniting all the data according with their gradient range (LEV: -2.5 to 2.5%; UH: 2.5 to 100% and DH: -100 to -2.5%), how the Authors can exclude that the observed pacing variation at a given gradient has not been affected by the other interludes occurred within a given gradient range?

Unfortunately, we do not have information relative to the participants' characteristics.

The reviewer is correct regarding the interpretation of findings based on the large range of gradients used to define the gradient categories. However, we do not think that a greater accuracy can be achieved in the current state of non-differential technology associated with the necessity to use an automated calculation of elevation in UM studies (please also refer to previous comment). Nonetheless, as the reviewer accurately indicated, this is an observational and descriptive study, and we do not make comments regarding the generalisation of our findings to all hilly UM events. Still, there is currently, no systematic description of pacing in UM of similar distances, despite the relatively large amount of publications linking physiological, neuromuscular, biomechanical, nutritional, psychological findings to performance and pacing during such events.

Regarding the environmental conditions, we have provided additional details. These conditions are mild, with no rain, and therefore we do not expect they had any substantial effect on pacing and pacing strategies. Please also refer to similar comment from Reviewer 1.

> **Paragraph #14.** (…) Weather conditions on race day were dry (no precipitations), the temperatures ranged 14.9-22.8 ℃ at the start point, 10.2-29.1 ℃ at the 103-km checkpoint, and 11.4-26.8 ℃ at the finish line, respectively.

Paragraph 16, lines 213-216: Although using the average running speed as index of running performance permits to compare the two different groups (173-km and 103-km participants), have the Authors also tried to correlate the pacing characteristics with the finishing time in the two groups separately before concluding that there is no correlation between these characteristics and endurance performance?

These results are readily provided in Table 1, reproduced below.

| | All participants | | 103-km | | 173-km | |
|---|---|---|---|---|---|---|
| | r | P | r | p | r | p |
| **Speed variability** | -0.23 | 0.42 | -0.24 | 0.70 | -0.25 | 0.48 |
| **Speed loss** | -0.24 | 0.39 | -0.43 | 0.47 | 0.00 | 1.00 |
| **Total time stopped** | -0.35 | 0.21 | -0.15 | 0.81 | -0.45 | 0.20 |

Abstract, lines 25-28: This sentence is unclear: how positive pacing could have characterized all gradients if a speed reserve has been observed in LEV and UH?

The two are not mutually exclusive since the "positive pacing" is a general observation (considering the entire event), while the "speed reserve" is a local observation (looking at the end of the event in isolation).

Abstract, lines 28-30: This conclusion does not seem valid since significant correlations between overall performance and pacing characteristics have been previously observed in 161-km mountain ultramarathons (Hoffman 2014).

We respectfully partially disagree with the reviewer since the study from Hoffmann (2014) focused solely on winners and top-5 finishers of a mountain UM (D+ 4737 / D- 7001 m). In the current study, the cohort of participants had a large range of performance levels (representative of the entire participant group of the event), and we consider the profile of the event to be merely "hilly" with 3,000 m of elevation gain and loss. Therefore, we are confident to indicate that the absence of correlation between characteristics of speed and performance is novel.

## Comments for the author

As I previously stated, this article is certainly interesting and it provides insight into some not well studied aspects of exercise science through a systematic analysis of pacing in a trail ultramarathon running event. However, there are some parts that definitely need to be corrected and/or rewritten in order to make it publishable.

We thank the reviewer once again for their very valuable comments, and for the opportunity to improve our article.

---

## Round 0.3 · accepted · Accept

· Academic Editor

Accept

Thank you for your careful efforts to deal with the second round of comments.